# Micro-Osteoperforations Induce TNF-α Expression and Accelerate Orthodontic Tooth Movement via TNF-α-Responsive Stromal Cells

**DOI:** 10.3390/ijms23062968

**Published:** 2022-03-09

**Authors:** Ria Kinjo, Hideki Kitaura, Saika Ogawa, Fumitoshi Ohori, Takahiro Noguchi, Aseel Marahleh, Yasuhiko Nara, Adya Pramusita, Jinghan Ma, Kayoko Kanou, Itaru Mizoguchi

**Affiliations:** Division of Orthodontics and Dentofacial Orthopedics, Graduate School of Dentistry, Tohoku University, 4-1, Seiryo-machi, Aoba-ku, Sendai 980-8575, Miyagi, Japan; ria.kinjou.p5@dc.tohoku.ac.jp (R.K.); saika.ogawa.a4@tohoku.ac.jp (S.O.); fumitoshi.ohori.t3@dc.tohoku.ac.jp (F.O.); takahiro.noguchi.r4@dc.tohoku.ac.jp (T.N.); marahleh.aseel.mahmoud.t6@dc.tohoku.ac.jp (A.M.); yasuhiko.nara.q6@dc.tohoku.ac.jp (Y.N.); adya.pramusita.q6@dc.tohoku.ac.jp (A.P.); ma.jinghan.s1@dc.tohoku.ac.jp (J.M.); kanou.kayoko.s7@dc.tohoku.ac.jp (K.K.); mizo@tohoku.ac.jp (I.M.)

**Keywords:** micro-osteoperforations, TNF-α, orthodontic tooth movement, osteoclast

## Abstract

Micro-osteoperforations (MOPs) have been reported to accelerate orthodontic tooth movement (OTM), and tumor necrosis factor (TNF)-α has been reported to play a crucial role in OTM. In this report, the influence of MOPs during OTM was analyzed. We evaluated the expression of TNF-α with and without MOPs by RT-PCR analysis. A Ni-Ti closed coil spring was fixed between the maxillary left first molar and the incisors as an OTM mouse model to move the first molar in the mesial direction. MOPs were prepared on the lingual side and mesial side of the upper first molars. Furthermore, to investigate the target cell of TNF-α for osteoclast formation during OTM with MOPs in vivo, we created four types of chimeric mice in which bone marrow of wild-type (WT) or TNF receptor 1- and 2-deficient mice (KO) was transplanted into lethally irradiated WT or KO mice. The results showed that MOPs increased TNF-α expression, the distance of tooth movement and osteoclast formation significantly. Furthermore, mice with TNF-α-responsive stromal cells showed a significant increase in tooth movement and number of osteoclasts by MOPs. We conclude that MOPs increase TNF-α expression, and tooth movement is dependent on TNF-α-responsive stromal cells.

## 1. Introduction

A key concern for orthodontics is the lengthy duration of orthodontic treatment. Therefore, shortening the duration of orthodontic treatment is an active research area. Many methods have been tested to accelerate tooth movement, including biological, mechanical, physical and surgical approaches [1]. Surgical procedures are the most consistent and achieve the desired results. Surgical methods that accelerate orthodontic tooth movement are based on the regional acceleratory phenomenon (RAP), which involves an enhanced remodeling process in response to harmful stimuli [2].

Various techniques have been developed to accelerate orthodontic tooth movement (OTM) to coordinate these biological processes. In 1959, a method was introduced to accelerate OTM by cutting the cortical bone between the teeth, leaving minor penetrations in the medullary bone after exposing the buccal and lingual alveolar bone with a full-thickness flap. A subapical horizontal cut that connects the interdental cuts is an osteotomy-style approach that penetrates the entire thickness of the alveoli [3]. This cut was introduced to corticotomy in 2008, with alveolar grafting and flap elevation, as a new technique for accelerating orthodontic bone formation [4]. Additionally, a less invasive cortical incision was also introduced. In these methods, a surgical blade was inserted into the alveolar bone horizontal and interdentally without raising the flap [5]. In addition, minimal invasive piezoelectric surgery has been introduced with a piezo-surgical knife rather than using a surgical blade [6]. Recently, cortical incisions were performed using surgical lasers. Piezocision and laser-assisted flapless corticotomy via the Er:YAG laser accelerated canine retraction in a randomized controlled trial [7]. It has been reported that cortical incisions with the Er-Cr:YSGG laser accelerated orthodontic tooth movement in a rabbit OTM model [8]. In a recent study, piezocision and the Er:YAG laser incision were successful in accelerating tooth movement in a rat OTM model [9]. These surgical methods for accelerating OTM are based on local acceleration phenomena.

Perforations of the alveolar bone are a surgical method that is termed micro-osteoperforations (MOPs), and MOPs can be used to accelerate OTM [10]. Microscopic tissue reactions from applying this procedure in animal studies have been examined, and the results reveal that MOPs showed RAP [11]. Several studies introduced MOPs with clinical replication in humans [10,12,13]. The reports suggest that MOPs accelerate OTM effectively. Perforations to the alveolar bone around the teeth have also been reported to accelerate OTM in experimental animal models [11,14,15,16,17]. Perforation procedures of the alveolar bone were found to induce expression of inflammatory cytokines because of enhanced osteoclast formation [11]. Thus, this procedure enhances bone remodeling and tooth movement during OTM. However, despite its proven effectiveness, the onset mechanism that accelerates OTM via MOP is poorly understood.

Osteoclasts derived from hematopoietic stem cells act a role in bone resorption and bone remodeling [18]. There are two essential factors for osteoclast formation: receptor activator of nuclear factor κB ligand (RANKL) and macrophage colony-stimulating factor (M-CSF) [19]. Moreover, tumor necrosis factor (TNF)-α, which is a pro-inflammatory cytokine, may also induce osteoclast differentiation [20,21,22,23] and may induce osteoclasts in bone erosive diseases as well [24].

OTM is dependent on alveolar bone remodeling by compression-force-associated osteoclasts and tension-force-associated osteoblasts, which results in bone resorption and bone formation, respectively [25,26]. The orthodontic force, induces cytokines, growth factors and neurotransmitters, which, when added to the compressed side in the periodontal ligament, induces osteoclasts, and the tooth moves as the osteoclasts resorb the bone.

Several reports showed that the compression force in OTM induces expression of TNF-α [27,28,29,30,31,32,33,34]. In our previous study, we also found that TNF-α plays important roles in an OTM mouse model by using both TNF receptor 1- and TNF receptor 2-deficient (KO) mice. Here, the distance the tooth moved was found to be shorter than that of the wild-type (WT) mice [35,36,37].

In our previous study, four kinds of chimeric mice were generated by a bone marrow transplantation method using WT and KO mice: (i) mice with stromal cells and macrophages bearing TNF receptors; (ii) mice with only stromal cells bearing TNF receptors; (iii) mice with only macrophages bearing TNF receptors; and (iv) mice with stromal cells and macrophages that are deficient in TNF receptors. Osteoclast formation was evaluated after injection of TNF-α into the supracalvariae of these chimeric mice. The results indicated that stromal cells contribute more to osteoclast formation than macrophages, although both macrophages and stromal cells were found to be direct targets of TNF-α [22,23]. Furthermore, the study showed that chimeric mice can be used to examine the type of cell that TNF-α targets for osteoclast formation during OTM. The results of this study suggested that the response of stromal cells to TNF-α is a crucial factor for osteoclast formation and bone resorption in OTM [37]. However, the effects of MOPs on TNF-α expression during OTM remain unknown. In this study, we analyzed TNF-α expression during experimental tooth movement with MOPs. We also investigated whether tooth movement is accelerated by MOPs-induced TNF-α using mice that are deficient in the expression of TNF receptors. Furthermore, chimeric mice were used to investigate the contribution of each TNF-α target cell type in vivo for tooth movement and osteoclast formation in OTM with MOPs.

## 2. Results

### 2.1. MOPs Increase Tooth Movement and Osteoclast Formation

After the mice were anesthetized, the mice were fixed in a supine position. Next, the mouth was kept open by pulling the lower incisors caudally. The left corner of the mouth was gently stretched laterally to access the upper left first molar. We made two points for MOPs. One MOP was placed about 1.0 mm mesial to the maxillary left first molar. The other MOP was placed about 1.0 mm palatal to the maxillary left first molar (Figure 1A). These MOPs were created by inserting a 0.5-mm diameter round steel bur in the alveolar bone with a slow-speed handpiece. Perforations were approximately 0.5 mm in diameter and 0.25 mm in depth. Hemostasis was achieved by using cotton pellets after the MOPs procedure. To achieve OTM under anesthesia, a Ni-Ti closed coil spring (Tomy, Fukushima, Japan) was fixed to the alveolar bone of the maxillary incisors and the left maxillary first molar. The left maxillary first molar was mesially moved. The appliance was tied to the upper first molar and incisor with a 0.1-mm diameter stainless steel wire. Based on the manufacturer’s instructions, the appliance was adjusted to 10 g, as described previously [35,37] (Figure 1B). Four mice were used in each group. The distance between the upper first and second molars was measured by impressing the teeth with Examixfine injection type material (GC Co., Tokyo, Japan) after OTM. The samples were immersed in 4% paraformaldehyde for fixation after the impression process. The distance the teeth moved was assessed using a stereoscopic microscope (VH-7000; Keyence, Osaka, Japan), as described previously [35,37] (Figure 1B).

After six days of mechanical loading, no significant difference between no MOPs and MOPs was observed for the WT mice (Figure 2A,B). On Day 12, tooth movement was greater for the WT mice with MOPs than that of the WT mice without MOPs (Figure 2A,B). Furthermore, tartrate-resistant acid phosphatase (TRAP) staining was performed on sections from the mesial side of the distobuccal root of the left maxillary first molars. Osteoclasts were detected along the alveolar bone on the pressure side of the tooth in the WT mice, and a similar number of osteoclasts were detected in both of the WT mice groups on Day 6 (Figure 2C,D). Significantly more osteoclasts were observed on the alveolar in the pressure side of the tooth on Day 12 than on Day 6, and the number of osteoclasts detected in the WT mice with MOPs on Day 12 was significantly greater than the WT mice without MOPs (Figure 2C,D). To evaluate the effect of MOPs, we examined TRAP-stained transverse sections of a tooth on the opposite side. There was no significant difference between the mice groups with and without MOPs on both Day 6 and Day 12 (Figure 2E,F).

### 2.2. MOPs Induce TNF-α Expression

We showed previously that in the absence of a TNF-α response, osteoclast formation and tooth movement in a mouse model decreased [35,36,37]. Therefore, we evaluated whether MOPs induced TNF-α expression. To evaluate TNF-α mRNA expression around the left maxillary first molar after MOPs, alveolar bones were excised from the maxilla around the left maxillary first molar after 0, 6 and 12 days of MOPs. TNF-α mRNA expression was evaluated. WT mice with MOPs showed a significant increase in TNF-α mRNA level when compared with that of WT mice without MOPs on Day 0 (Figure 3).

### 2.3. MOPs Induce Tooth Movement and Osteoclast Formation Is Dependent on TNF-α

On Day 6, there was no significant difference between the WT and KO mice. Moreover, there was also no significant difference in tooth movement between the WT and KO mice with or without MOPs (Figure 4A,B). On Day 12, tooth movement was greater in the WT mice than that of the KO mice. Furthermore, tooth movement in the WT mice with MOPs was greater than that observed for the WT mice without MOPs. However, there was no significant difference between the KO mice with or without MOPs after 12 days (Figure 4A,B).

TRAP staining of sections from the mesial side of the distobuccal root of the maxillary left first molars was performed. Osteoclasts were observed along the alveolar bone on the pressure side of the tooth in the WT mice, and a similar number of osteoclasts were detected in the KO mice on Day 6 (Figure 4C,D). The number of osteoclasts in the WT mice with MOPs was also similar to that observed in the KO mice with MOPs on Day 6. No significant difference in the osteoclast numbers between the WT and KO mice with or without MOPs on Day 6 was observed (Figure 4C,D). In contrast, many osteoclasts were observed along the alveolar bone on the pressure side of the tooth in the WT mice, whereas fewer osteoclasts were observed in the KO mice on Day 12. There were more osteoclasts in the WT mice than in the KO mice on Day 12 in OTM (Figure 4C,D). The number of osteoclasts in the WT mice with MOPs was much greater than observed in the WT mice without MOPs on Day 12. (Figure 4C,D). However, the number of TRAP-positive cells in the KO mice with MOPs was similar in number to the TRAP-positive cells in the KO mice on Day 12. There was no significant difference in osteoclast numbers between the KO and KO mice with MOPs on Day 12 (Figure 4C,D).

### 2.4. MOPs Increased Osteoclast Formation and Tooth Movement In Vivo via TNF-α-Responsive Stromal Cells

MOPs affected the OTM distance and osteoclast formation in OTM. Four kinds of chimeric mice were generated to analyze which cell type is targeted by TNF-α in order to assess their contribution to MOP-assisted OTM. The chimeric mice prepared were: (i) bone marrow cells of WT transplanted into lethally gamma-irradiated WT mice (WT > WT); (ii) bone marrow cells of WT transplanted into lethally gamma-irradiated KO mice (WT > KO); (iii) bone marrow cells of KO transplanted into lethally gamma-irradiated WT mice (KO > WT); and (iv) bone marrow cells of KO transplanted into lethally gamma-irradiated KO mice (KO > KO). After 12 days of tooth movement with or without MOPs, the distance of OTM was evaluated, and osteoclasts along the alveolar bone on the compression side were counted as osteoclasts. The distance of OTM in the WT > KO and KO > KO chimeric mice was significantly shorter than the distance of OTM in the WT > WT and KO > WT chimeric mice (Figure 5A,B). No significant difference was observed between the distance of OTM in the WT > KO and KO > KO chimeric mice or between the WT > WT and KO > WT chimeric mice. Furthermore, the distances of OTM in the WT > WT and KO > WT mice with MOPs were significantly longer than the distances of OTM in the WT > WT and KO > WT mice without MOPs, respectively (Figure 5A,B). However, no significant difference was observed between the distance of OTM between the WT > KO with MOPs and WT > KO mice without MOPs or between the KO > KO with MOPs and KO > KO mice without MOPs (Figure 5A,B).

The osteoclasts number along the alveolar bone on the compression side in the WT > KO and KO > KO mice was also significantly lower than the number of TRAP-positive cells in the WT > WT and KO > WT mice (Figure 6C,D). Moreover, no significant difference was observed in the number of osteoclasts along the alveolar bone on the compression side between the WT > KO and KO > KO mice or between the WT > WT and KO > WT mice (Figure 5C,D). Furthermore, the number of osteoclasts along the alveolar bone in the compression side in the WT > WT and KO > WT mice with MOPs was significantly higher than the number of osteoclasts in the WT > WT and KO > WT mice without MOPs, respectively (Figure 5C,D). However, no significant difference was observed in the number of osteoclasts on the alveolar bone in the compression side between the WT > KO with MOPs and WT > KO mice without MOPs or between the KO > KO with MOPs and KO > KO mice without MOPs (Figure 5C,D).

### 2.5. TNF-α Induces RANKL Expression in Stromal Cells in a Dose-Dependent Manner

We evaluated the possibility that TNF-α-induced RANKL expression is dose dependent. Bone marrow stromal cells were cultured with several doses of TNF-α (0, 1, 10 and 100 ng/mL). Total RNA was obtained from adherent cells after 4 days of culturing. The increasing RANKL expression in stromal cells was observed to be dependent on the TNF-α dose (Figure 6A). 

### 2.6. TNF-α Enhances Osteoclast Formation in a Chimeric Co-Culture System, Which Used KO Osteoclast Precursors and WT Stromal Cells

To evaluate whether TNF-α-induced expression of RANKL in stromal cells and enhanced osteoclast formation are dependent on the dose of TNF-α in a co-culture, we used KO osteoclast precursors to eliminate the direct effect of TNF-α on osteoclast precursors in osteoclast formation. We cultured WT stromal cells and KO osteoclast precursors with several doses of TNF-α (0, 1, 10 and 100 ng/mL), 10^−6^ M prostaglandin E2 and 10^−8^ M 1,25(OH)_2_D_3_. We counted the osteoclast number in the co-culture. It showed that the osteoclast number increased significantly in a TNF-α dose-dependent manner (Figure 6B,C), and the percentage of resorption pits also increased significantly in a TNF-α dose-dependent manner (Figure 6D,E).

## 3. Discussion

MOPs have been reported to accelerate OTM [10,12,13], and TNF-α has been reported to play a significant role in OTM [35,36,37]. However, the effect of MOPs on TNF-α expression during OTM remains unknown. In this study, the effect of MOPs on TNF-α expression and OTM was investigated. A Ni-Ti closed coil spring was placed between the maxillary left first molar and the maxillary incisors as an OTM mouse model to move the first molar in the mesial direction of C57BL/6J mice for a maximum of 12 days. MOPs were made on the palatal side and mesial side of the upper first molars by using a round bur. Tooth movement and histological effects with and without MOPs were assessed. Furthermore, to investigate the target cell of TNF-α for osteoclast formation during OTM with MOPs in vivo, we created WT > WT, KO > WT, WT > KO and KO > KO chimeric mice. MOPs increased TNF-α expression and the distance of tooth movement significantly. Furthermore, the WT > WT and KO > WT chimeric mice groups with MOPs significantly increased the distance of tooth movement and number of osteoclasts when compared with the results without MOPs. In the WT > KO and KO > KO mice, MOPS did not cause an increase in tooth movement and the number of osteoclasts. The results suggest that MOPs accelerate tooth movement, and this movement is dependent on the response of stromal cells to TNF-α. This is the first report describing the effect of MOPs using KO mice and specific chimeric mice.

MOPs have been reported to induce inflammatory cytokines, TNF-α, IL-1, IL-3, IL-6, IL-11 and IL-18, in rat periodontal ligament cells [11]. Moreover, it has been reported that MOPs also increase OTM and enhance expression of inflammatory markers, including TNF-α, in human gingival crevicular fluid [10]. These results suggest that these inflammatory cytokines induced by MOPs enhance osteoclast formation and bone resorption, which accelerates OTM. However, these results do not explain how MOP-induced inflammatory cytokines enhance OTM. Several previous reports showed that TNF-α is induced by orthodontic force, and TNF-α plays a significant role in osteoclast formation and bone resorption in OTM [27,28,29,30,31,32,33,34]. Our previous studies also showed that TNF-α plays an important role in a mouse OTM model with an shorter average tooth movement distance observed for KO mice than that of WT mice [35,36]. An immunostaining study reported the expression of TNF-α on the compression side of the periodontal ligament in a rat OTM model which was enhanced by MOPs [16]. In this study, expression of TNF-α mRNA in mice was increased by MOPs. The results suggest that MOPs-induced TNF-α expression may be linked with MOPs-induced acceleration of OTM. In our previous study, the distance of tooth movement and osteoclast formation in WT and KO mice during an experimental period were evaluated. A large amount of tooth movement occurred on Day 2 in both groups. From Days 4 to 8, large tooth movement was not observed in either group. On Day 10, the distance of tooth movement increased in the WT mice, but not in the KO mice. On Day 12, tooth movement increased in both groups, but the distance in the KO mice was significantly lower than that in the WT mice. The number of osteoclasts increased gradually in both of mice. On Day 6, there was no significant difference in the number of osteoclasts in both groups. However, the KO mice had significantly fewer osteoclast number than the WT mice on Day 12 [35]. Therefore, in the present study, we evaluated both Day 6 and Day 12: Day 6 was evaluated as a day when there is no difference, and Day 12 as the day when there is a difference between the WT and KO groups. In the present study, on Day 6, there was no significant difference of tooth movement and osteoclast formation with MOPs between the WT and KO mice; however on Day 12, tooth movement and osteoclast formation were greater in the WT mice with MOPs than the WT mice without MOPs. Conversely, there is no significant difference between the KO mice with MOPs and without MOPs. We found that the early stage of OTM is not affect by MOPs-induced TNF-α expression. We evaluated histological sections from the side opposite to the MOPs to evaluate the effect of MOPs on the surrounding area’s osteoclast formation. Here, no difference in osteoclast formation for the mice with and without MOPs after Day 6 and Day 12 was observed. These results suggest that MOPs affected only the local area near the site of MOPs.

A previous study investigated the contribution of macrophages and stromal cells to osteoclast formation induced by TNF-α in vivo [22,23]. These reports showed that stromal cells contributed more to osteoclast formation induced by TNF-α than macrophages. The KO > WT chimeric mice showed an increase in osteoclast formation when compared with the WT > KO chimeric mice [22,23]. Thus, these studies concluded that stromal cells induced by TNF-α contributed more to osteoclast formation in vivo than that of macrophages. In our previous study, four chimeric mice were used to determine the in vivo contribution of each cell type to OTM. The number of osteoclasts and distance of OTM were greater in the WT > WT and KO > WT mice than those in the WT > KO and KO > KO mice. In addition, the OTM distance and osteoclast numbers for the WT > KO mice were found to not be significantly different to those observed for the KO > KO mice. The increase in OTM distance and osteoclast numbers for the KO > WT mice was similar to those values for the WT > WT mice. Moreover, the WT > KO mice showed significantly shorter OTM distances and a lower osteoclast number than the WT > WT mice [37]. The findings showed that the response of stromal cells to TNF-α is a crucial factor for osteoclast formation and bone resorption in OTM. In the present study, the amount of OTM in the WT > WT and KO > WT mice with MOPs was higher than the amount in similar chimeric mice without MOPs. However, no significant difference was observed between the WT > KO mice with MOPs and the WT > KO mice without MOPs or between the KO > KO mice with MOPs and the KO > KO mice without MOPs. The results indicate that MOPs induce TNF-α expression, and the response of stromal cells to TNF-α is important for MOPs-enhanced OTM and osteoclast formation. 

Several studies showed that PDL cells expressed TNFRs [38,39,40,41]. Furthermore, it has been reported that the expression levels of TNFRs in CD146 positive PDL cells were higher than those in CD146 negative PDL cells, and TNF-α was found to have higher level of biological effect in CD146 positive PDL cells than in CD146 negative PDL cells [38]. However, there is no report of the immunohistological examination for expression of TNFRs in periodontal membrane. It is still unclear whether PDL cells express TNFRs. Therefore, we examined whether TNFRs express in PDL cells by immunohistochemistry. TNFR1 and TNFR2 expressed in PDL cells in the WT mice but no TNFRs expressed in the KO mice (Appendix A). The results indicated that TNFRs expressed as a protein level in PDL cells. Furthermore, we investigated TNFR expression on bone marrow cells in each chimeric mouse by fluorescent-activated cell sorting (FACS). Bone marrow macrophages of WT > WT and WT > KO expressed TNFR1 and TNFR2 but not KO > WT and KO > KO (Appendix A). The results suggest that at least the osteoclast precursors derived from hematopoietic cells expressed TNFRs in PDL cells in the WT > WT and WT > KO mice. However, it is still not clear exactly which kind of PDL cells expressed. Further studies are need to elucidate these points.

Several reports have demonstrated that TNF-α enhances RANKL expression in stromal cells [22,23,42,43,44,45]. In the present study, we confirmed that TNF-α enhanced the RANKL expression in stromal cells in a dose-dependent manner. Furthermore, we also evaluated whether TNF-α enhanced the osteoclast formation in a co-culture system consisting of TNF-α-activated stromal cells and TNFRs and whether KO osteoclast precursors might eliminate the direct TNF-α effect on osteoclast precursors as well as the TNF-α effect on stromal cells RANKL expression. The results demonstrated that TNF-α enhanced the RANKL expression in stromal cells. Furthermore, TNF-α enhanced osteoclastogenesis and bone resorption in the co-culture using KO osteoclast precursors in a dose-dependent manner. The results indicate that increasing the amount of TNF-α enhances RANKL expression in stromal cells, therefore enhancing osteoclast formation and bone resorption.

In a similar experiment, OTM in experimentally induced periodontitis in rats showed an increased rate of distance of OTM compared with OTM without periodontitis. Thus, periodontitis-induced bone loss was enhanced by OTM. It was concluded that OTM should be initiated after successful periodontal therapy [46]. Conversely, it has been reported that tooth movement was significantly decreased by experimentally induced periodontitis in rats [47]. It is unclear whether periodontal inflammation enhances OTM or not. Therefore, further study is needed to elucidate this point.

In the present study, MOPs enhanced tooth movement and osteoclast formation via TNF-α-responsive stromal cells. Furthermore, MOPs enhanced TNF-α expression in OTM. This is probably because TNF-α acts on stromal cells to increase RANKL expression. MOPs-enhanced TNF-α expression may increase the RANKL expression in stromal cells, and RANKL then enhances the osteoclast formation and tooth movement in OTM.

## 4. Materials and Methods

### 4.1. Mice

Nine- to ten-week-old C57BL6/J male mice were purchased from CLEA Japan (Tokyo, Japan), and KO mice (*Tnfrsf1a^tm1lmx^Tnfrsf1b^tm1lmx^*) were obtained from the Jackson Laboratory (Bar Harbor, ME, USA). The mice were housed under standard conditions (21–24 °C, 12/12-h light/dark cycle). A granular diet (Oriental Yeast, Tokyo, Japan) was used as feed for the mice to avoid excessive chewing power [35,37]. The protocols for all animal procedures were approved and reviewed by the Tohoku University of Science Animal Care and Use Committee.

### 4.2. RNA Preparation and Real-Time PCR Analysis

To evaluate TNF-α expression around the maxillary left first molar after MOPs, alveolar bones of the maxilla around the maxillary left first molar were extracted at 0, 3, 6 and 12 days after MOPs. To extract RNA, alveolar bone around the area of MOPs was dissected and frozen in liquid nitrogen after mice were sacrificed, and then frozen alveolar bones of left maxillae around the maxillary left first molar were crushed and homogenized by using a Micro Smash MS-100R (Tomy Seiko Co. Ltd., Tokyo, Japan). These samples were suspended in TRIzol Reagent (Invitrogen, Carlsbad, CA, USA), and then total RNA was isolated and purified from samples using RNeasy Mini Kit (Qiagen, Hilden, Germany), as described previously [48]. The amount of total RNA was evaluated with 260 nm absorbance. For cDNA synthesizing by SuperScript IV Reverse Transcriptase (Invitrogen), cDNA synthesized from two μg of total RNA was used. TB Green Premix Ex Taq II (Takara, Shiga, Japan) with the Thermal Cycler Dice Real Time System (Takara) were used for real-time PCR. PCR was performed as follows: initial denaturation stage (30 s at 95 °C), amplification stage (50 amplification cycles with each cycle composed of a denaturation step of 5 s at 95 °C and an annealing step of 30 s at 60 °C) and final dissociation stage (15 s at 95 °C, 30 s at 60 °C and 15 s at 95 °C). Glyceraldehyde 3-phosphate dehydrogenase (GAPDH) mRNA was used for normalizing expression levels. The following primers were used as described previously [49]: GAPDH, 5′-GGTGGAGCCAAAAGGGTCA-3′ and 5′-GGGGGCTAAGCAGTTGGT-3′; TNF-α, 5′- AGGCGGTGCTTGTTCCTCA-3′ and 5′-AGGCGAGAAGATGATCTGACTGCC-3′; and RANKL, 5′-CCTGAGGCCAGCCATTT-3′ and 5′-CTTGGCCCAGCCTCGAT-3′.

### 4.3. Procedure for Bone Marrow Transplantation

In our previous studies, we established a method for WT and KO mice bone marrow transplantation [22,23,37]. Mice were exposed to a lethal dose of gamma-irradiation (10 Gy). Subsequently, bone marrow cells (1 × 10^6^ cells in 100 μL PBS) from WT or KO mice were transplanted via the tail vein of the WT or KO mice. We generated four kinds of chimeric mice for investigating the contribution of TNF-α target cell types in vivo to MOPs-enhanced OTM and osteoclast formation. The chimeric mice: (i) WT > WT, in which WT bone marrow cells from WT were transplanted into a lethal dose of gamma-irradiated WT; (ii) WT > KO, in which WT bone marrow cells were transplanted into a lethal dose of gamma-irradiated KO; (iii) KO > WT, in which the bone marrow cells of KO were transplanted into a lethal dose of gamma-irradiated WT; and (iv) KO > KO, in which the bone marrow cells of KO were transplanted into a lethal dose of gamma-irradiated KO.

### 4.4. Sample Preparation for Histological Examination

After fixing, the samples were decalcified in 14% ethylenediaminetetraacetic acid (EDTA) for 30 days. The solution was changed twice a week. After decalcification, the samples were dehydrated by using graded ethanol and embedded in paraffin. Then, they were sectioned in the horizontal plane at 4 µm. Sections at 100, 140, 180, 220 and 260 µm apical to the maxillary left first molar bifurcation area were stained for TRAP activity to evaluate osteoclasts in every section. Fifty millimolar sodium tartrate, Fast Red Violet LB Salt (Sigma-Aldrich, St. Louis, MO, USA) and Naphthol-ASMX-phosphate (Sigma-Aldrich) were used for TRAP staining. Osteoclasts were defined as multinucleated TRAP-positive cells on the bone surface. They were counted by using light microscopy, as described previously [35,37]. The area of counting was the mesial side of the distobuccal root of the maxillary left first molar, and every section was counted. The mean values were calculated in all five sections, as described previously [35,37].

### 4.5. Procedure for Bone Marrow Stromal Cells

Bone marrow cells were obtained from the bone marrow cavity of the femora and tibia by flushing with PBS. After washing with PBS, the obtained bone marrow cells were cultured in Dulbecco’s Modified Eagle Medium (Sigma-Aldrich) with 10% FBS, 100 μg/mL streptomycin and 100 IU/mL penicillin G (Meiji Seika, Tokyo, Japan) for 2 weeks. After culturing, the suspended cells were washed thoroughly with PBS. The remaining adherent cells were detached by a trypsin-EDTA (Sigma-Aldrich). We used the cells as bone marrow stromal cells, as described previously [50].

### 4.6. Preparation of Osteoclast Precursors and Co-Culturing of Osteoclast Precursors from KO Mice and Stromal Cells from WT

The bone marrow cells of KO mice were obtained by flushing and cultured in α-MEM that included 10% FBS, 100 μg/mL streptomycin and 100 IU/mL penicillin G with 100 ng/mL M-CSF for 4 days. After PBS washing, the adherent cells were obtained by using trypsin-EDTA. The cells were used as osteoclast precursors. KO osteoclast precursors and WT bone marrow stromal cells were co-cultured with several doses of TNF-α (0, 1, 10 and 100 ng/mL), 10^–6^ M prostaglandin and 10^–8^ M 1,25(OH)_2_D_3_ (Sigma-Aldrich) in a 96-well plate and in Osteo Assay Plate 96 Wells (Corning life Sciences, Corning, NY, USA). The co-culture was maintained for 4 days and then terminated by fixation with 4% paraformaldehyde and stained with TRAP staining consisting of acetate buffer (pH 5.0), naphthol AS-MX phosphate and 50 mM sodium tartrate Fast Red Violet LB salt. Cells that were TRAP-positive and had two or more nuclei were considered to be osteoclasts. Formation of pits was evaluated after 5 days of culturing. 

### 4.7. Statistical Analysis

The Scheffe F test was performed to evaluate significance, and *p* values less than 0.05 were considered to be statistically significant. All quantitative results are presented as the mean ± standard deviation. 

## 5. Conclusions

MOPs increased OTM distance and TNF-α expression. The responsiveness of stromal cells to TNF-α is a crucial factor for osteoclast formation, and MOPs amplify this response, leading to increased movement distance during OTM (Figure 7). This work provides a theoretical base for applications in humans. However, many variables come into play if MOPs are to be practiced in humans, such as the number, location, and timing of MOPs. These factors should be taken into consideration for MOPs-induced TNF-α to act effectively on stromal cells in future experiments studying this application in humans.

## Figures and Tables

**Figure 1 ijms-23-02968-f001:**
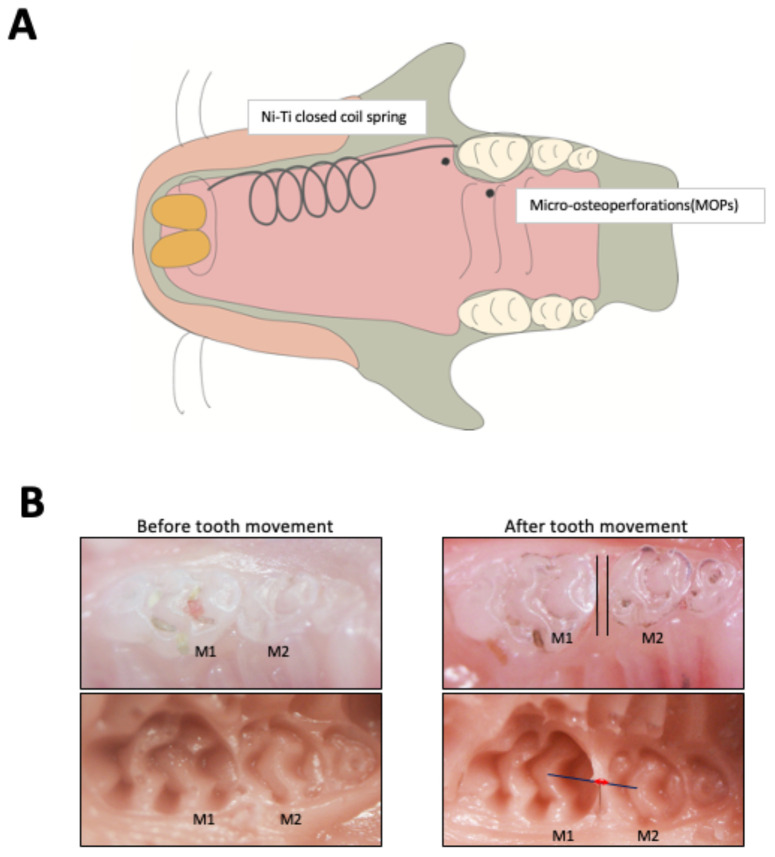
Orthodontic tooth movement and region of MOPs in mice: (**A**) Schematic of the appliance fixed between incisors and the first molar in mice and MOPs. One MOP (black dot) was placed about 1.0 mm mesial to the maxillary left first molar. The other MOP (black dot) was placed about 1.0 mm palatal to the maxillary left first molar. (**B**) Pictures of tooth movement and measurement of OTM in mice. The red double arrow between the maxillary left first molar (M1) and maxillary left second molar (M2) on the solid line connecting the central fossae of the two molars in silicone impressions was evaluated by stereoscopic microscopy.

**Figure 2 ijms-23-02968-f002:**
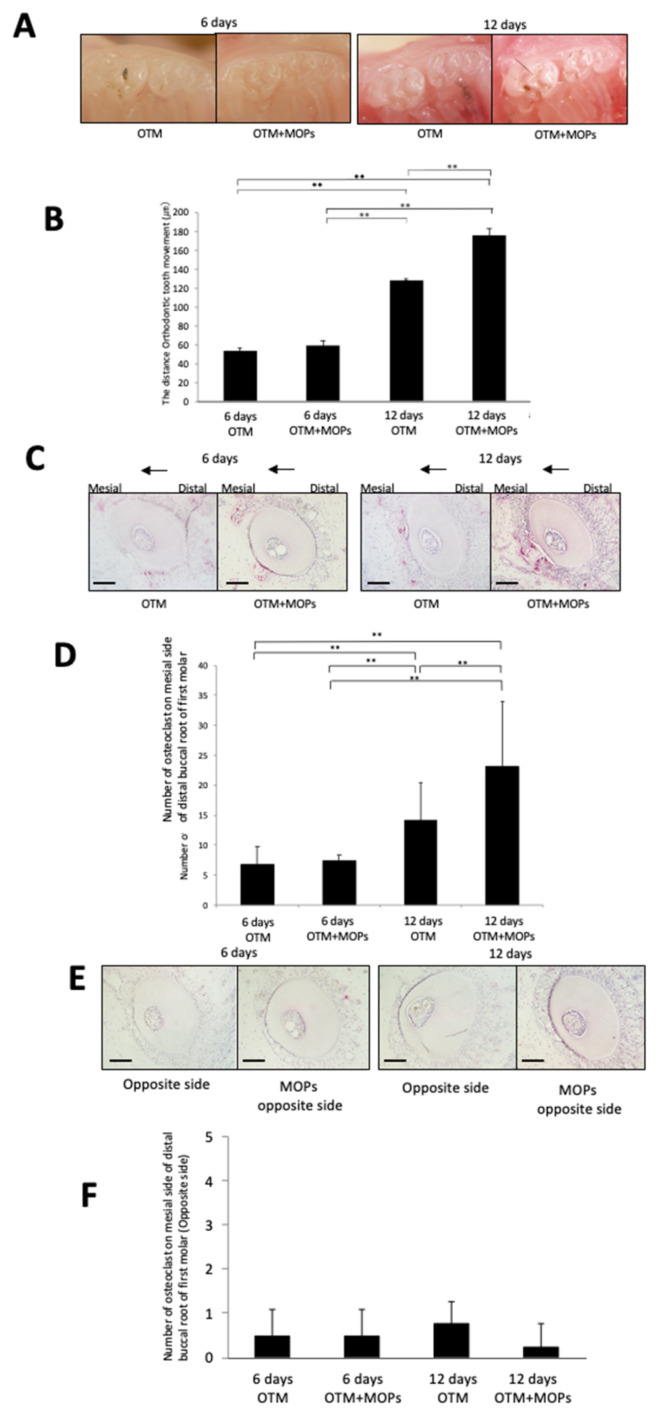
Effect of MOPs on orthodontic tooth movement and histology analysis of MOPs on orthodontic tooth movement: (**A**) Images of teeth after 0, 6 and 12 days of experimental loading with and without MOPs in WT mice. (**B**) Distance of tooth movement with and without MOPs in WT mice after 0, 6 and 12 days of experimental loading. (**C**) TRAP-stained histological sections of the distobuccal root of the maxillary left first molar after 6 and 12 days of experimental loading with and without MOPs in WT mice. Arrows indicate the direction of OTM. (**D**) Evaluation of the number of osteoclasts on the mesial side of the distobuccal upper-left first molar. (**E**) Histological examination was performed to evaluate the area affected by MOPs. TRAP-stained sections of the opposite side of WT mice with and without MOPs on Day 6 and Day 12. (**F**) The number of osteoclasts on the opposite side of WT mice with and without MOPs on Day 6 and Day 12. The results are presented as the mean ± standard deviation (*n* = 4). ** *p* < 0.01 indicate significant differences, which were analyzed by using the Scheffe test. Scale bars = 100 μm.

**Figure 3 ijms-23-02968-f003:**
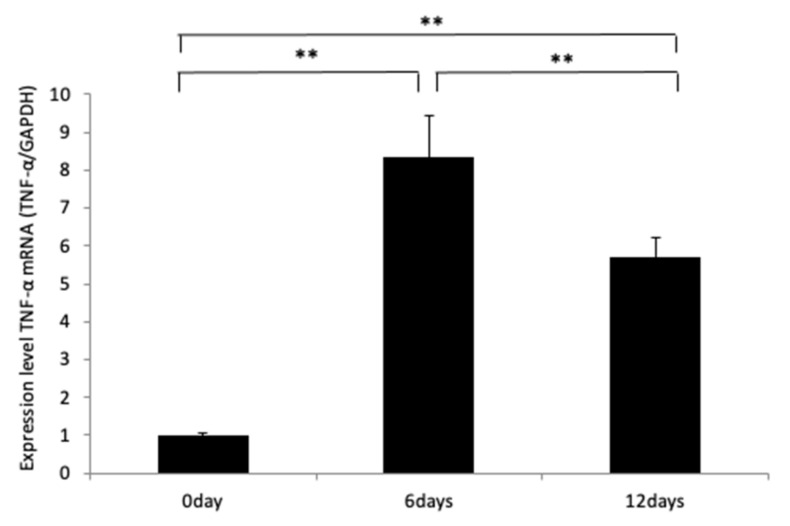
Time course of TNF-α expression by WT mice with MOPs. The alveolar bones of the left side maxillary first molar were excised after 0, 6 and 12 days with MOPs to evaluate the expression of TNF-α mRNA around the left maxillary first molar after MOPs. TNF-α mRNA expression was evaluated by real-time RT-PCR. The levels of TNF-α mRNA were normalized to the levels of GAPDH. The results are presented as the mean ± standard deviation (*n* = 4). ** *p* < 0.01 indicated significant differences, which were analyzed by using the Scheffe’s test.

**Figure 4 ijms-23-02968-f004:**
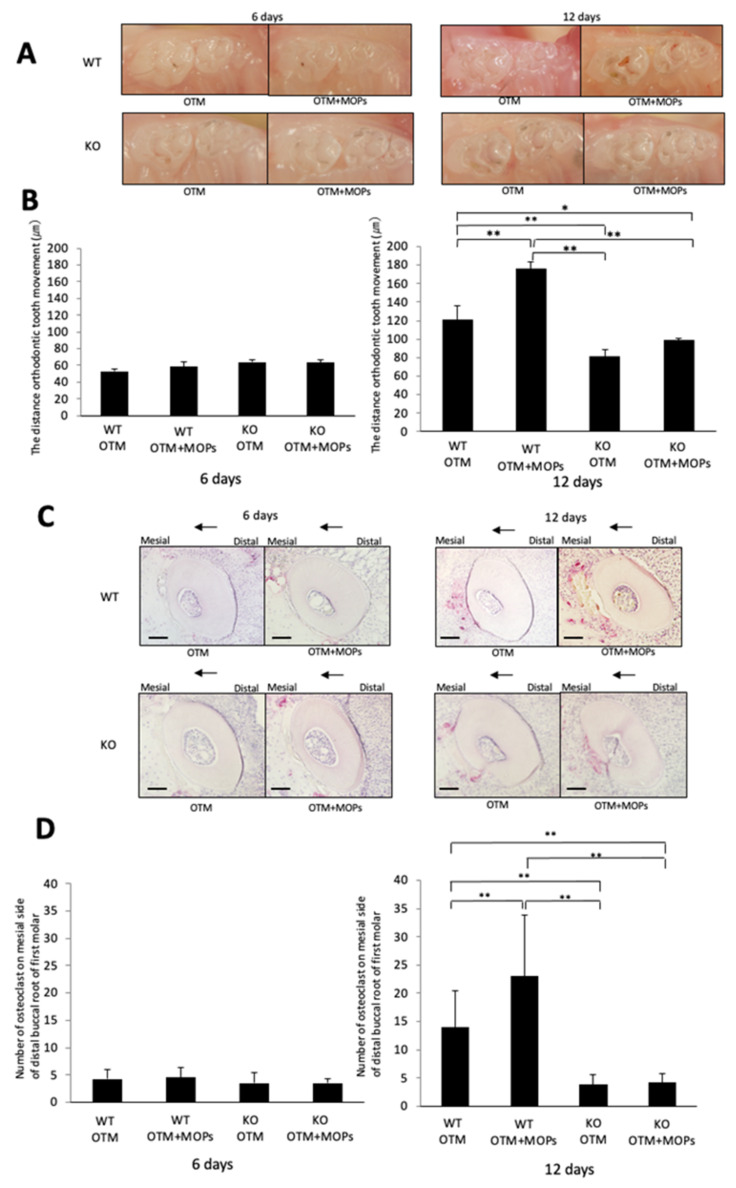
MOPs-induced tooth movement and osteoclast formation is dependent on TNF-α: (**A**) Images of teeth after 6 and 12 days of OTM in WT and KO mice with and without MOPs. (**B**) Measured distance of tooth movement in WT and KO mice with and without MOPs after 6 and 12 days of experimental loading. (**C**) TRAP-stained histological sections of the distobuccal root of the maxillary left first molar in WT and KO mice with and without MOPs after 6 and 12 days of experimental loading. (**D**) The TRAP-positive osteoclast cell number at the alveolar bone surface around the distobuccal root of the left maxillary first molar after 6 and 12 days of OTM for WT and KO mice with and without MOPs. The results are given as the mean ± standard deviation (*n* = 4). ** *p* < 0.01 and * *p* < 0.05 indicate significant differences, which were analyzed by using the Scheffe test. Scale bars = 100 μm.

**Figure 5 ijms-23-02968-f005:**
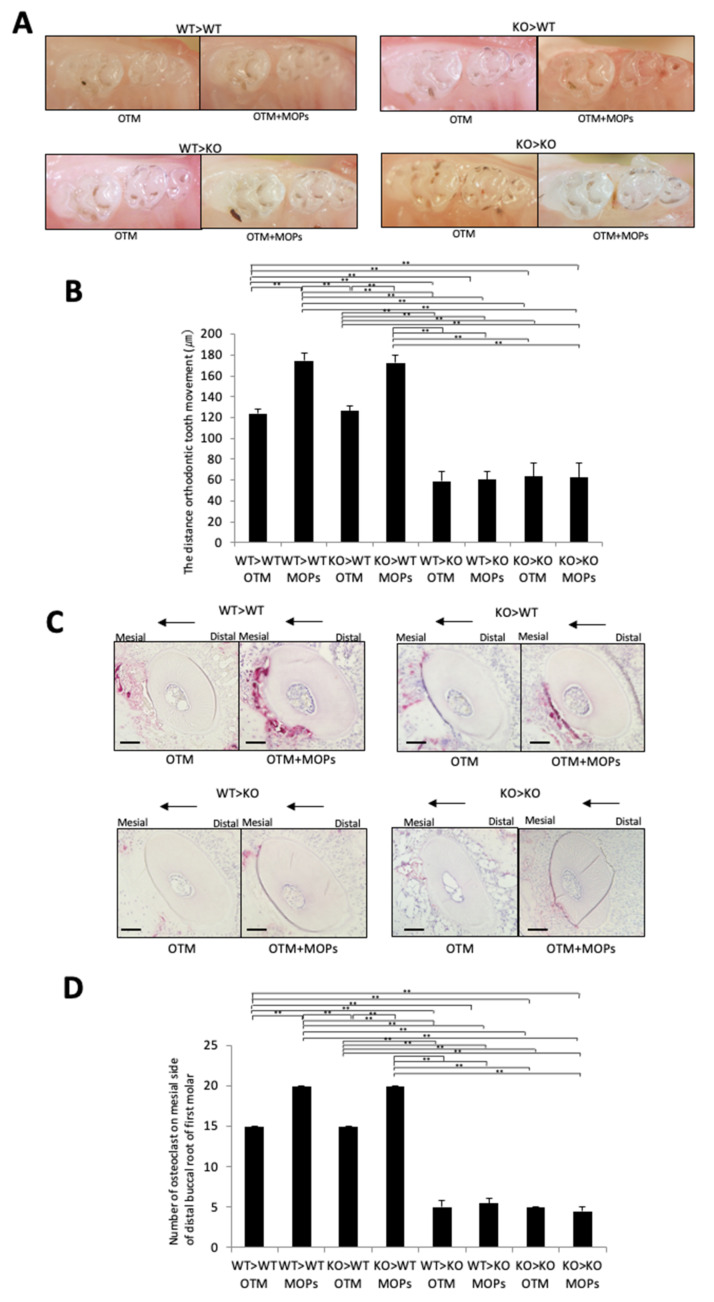
Tooth movement in chimeric WT and KO mice with or without MOPs after 12 days of OTM: (**A**) Images of tooth movements after 12 days of orthodontic force loading in chimeric WT and KO mice with and without MOPs. (**B**) Measured tooth movement distance in chimeric WT and KO mice with and without MOPs after 12 days of experimental loading. (**C**) TRAP-stained histological sections of the distobuccal root of the maxillary left first molar in chimeric WT and KO mice after 12 days of experimental loading. (**D**) The TRAP-positive osteoclast cell number along the alveolar bone surface around the distobuccal root of the left maxillary first molar after 12 days of OTM for chimeric WT and KO mice with and without MOPs. The results are given as the mean ± standard deviation (*n* = 4). ** *p* < 0.01 indicate significant differences, which were analyzed by using the Scheffe test. Scale bars = 100 μm.

**Figure 6 ijms-23-02968-f006:**
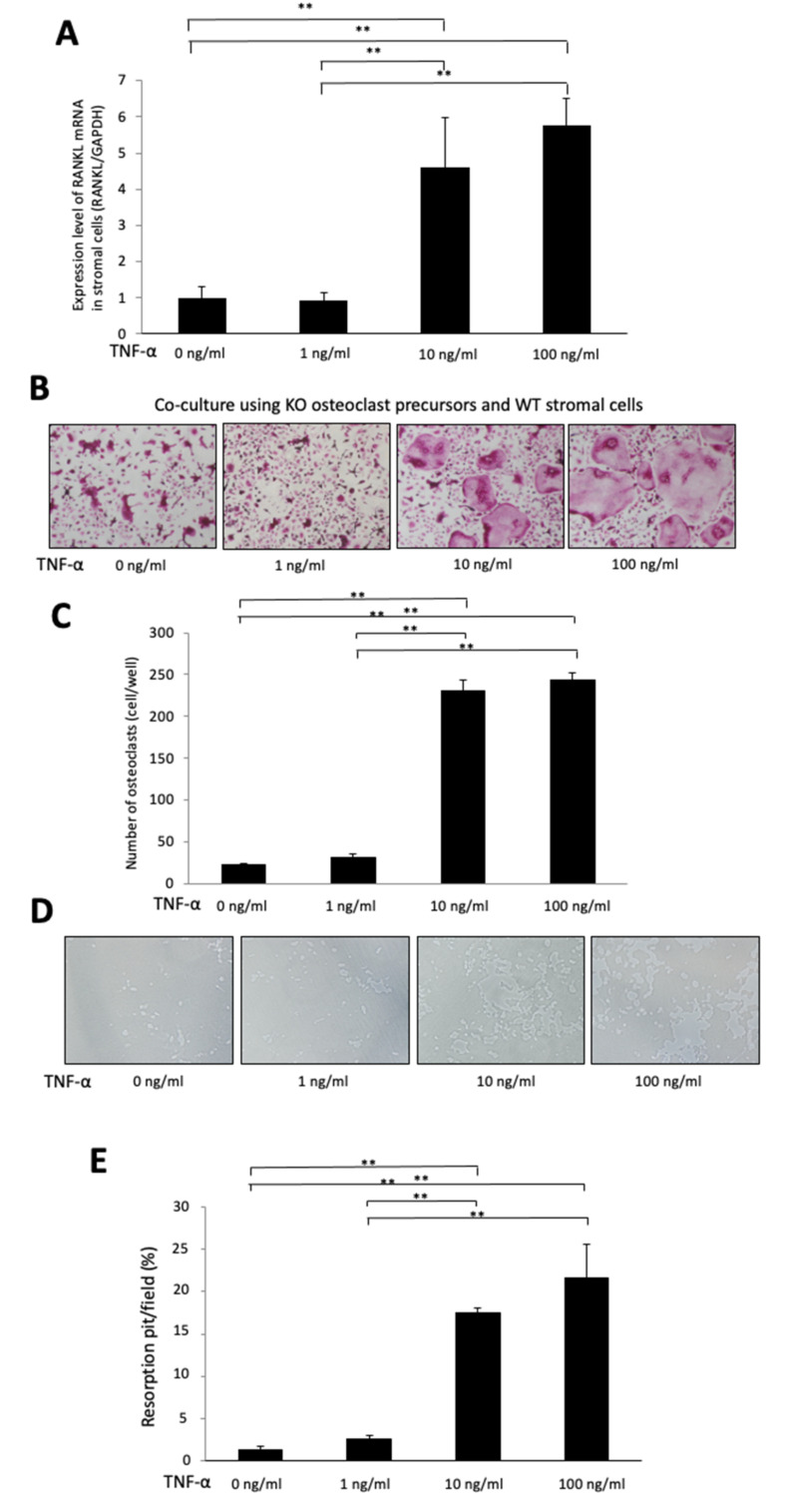
TNF-α induces RANKL expression in WT stromal cells and induced osteoclast formation in a co-culture using KO osteoclast precursors to eliminate the effect of TNF-α on osteoclast formation and WT stromal cells in a dose-dependent manner: (**A**) Real-time RT-PCR analysis of expression levels of RANKL mRNA in WT stromal cells. Total RNA was obtained from WT stromal cells cultured with TNF-α (0, 1, 10 and 100 ng/mL) for 3 days. (**B**) Images of TRAP-positive cells and (**C**) the number of TRAP-positive cells. (**D**) Images of the resorption pits and (**E**) the percentage of resorption pits in co-cultures of WT stromal cells and KO osteoclast precursors cultured with TNF-α (0, 1, 10 and 100 ng/mL) in the present of prostaglandin E2 and 1,25(OH)_2_D_3_ for 4 days. The results are presented as the mean ± standard deviation (*n* = 4). ** *p* < 0.01 indicate significant differences, which were analyzed by using the Scheffe test.

**Figure 7 ijms-23-02968-f007:**
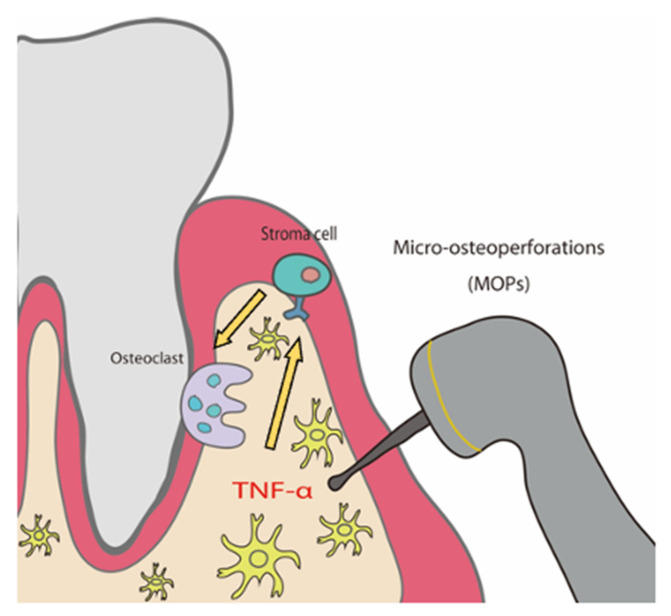
Schema of the mechanism of MOPs-enhanced osteoclast formation and MOPs-accelerated OTM.

## Data Availability

Data are available from the corresponding authors upon reasonable request.

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
