# Peer review of "Micro-Osteoperforations Induce TNF-α Expression and Accelerate Orthodontic Tooth Movement via TNF-α-Responsive Stromal Cells"

_ijms, 2022, doi:10.3390/ijms23062968_

Round 1

Reviewer 1 Report

In this “revised” manuscript, the authors quite sincerely responded to the reviewer’s comments, and performed immunohistochemistry for TNF receptors. The results are fascinating and excellent from the point of optimizing the immune-staining, meaning of the results, and patience for performing the time-consuming experiments.

These novel findings make it possible to visible TNFR-positive cells in periodontal ligament. Furthermore, flow-cytometry for TNFR clearly demonstrated the selective expression of TNFR on osteoclast precursors.

These revision is quite excellent, and the reviewer felt that the manuscript is now worth publishing.

Reviewer 2 Report

Dear authors,

As I have reviewed this paper before you have satisfied most of my requirements regarding it.

My only recommendations for you in this stage would be to introduce in the Discussion section a paragraph regarding the importance of certain salivary MMPs and ILs related to periodontal disease during orthodontic treatment. I think it might be an interesting comparison.

Also the potential future clinical importance of your findings should be highlighted in a better way in the Conclusions section.

Please receive my best regards!